# Simultaneous Denitrification and Bio-Methanol Production for Sustainable Operation of Biogas Plants

**I-Tae Kim** 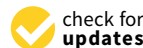

Division of Environment and Plant Engineering, Korea Institute of Civil Engineering and Building Technology 283, Goyang-daero, Ilsanseo-gu, Goyang-si, Gyeonggi-do 10223, Korea; itkim@kict.re.kr; Tel.: +82-31-9100885

**Abstract:** This study was conducted to secure the sustainability of biogas plants for generating resources from food waste (FW) leachates, which are prohibited from marine dumping and have been obligated to be completely treated on land since 2013 in South Korea. The aim of this study is to reduce the nitrogen load of the treatment process while producing bio-methanol using digested FW leachate diverted into wastewater treatment plants. By using biogas in conditions where methylobacter (*M. marinus* 88.2%) with strong tolerance to highly chlorinated FW leachate dominated, 3.82 mM of methanol production and 56.1% of total nitrogen (TN) removal were possible. Therefore, the proposed method can contribute to improving the treatment efficiency by accommodating twice the current carried-in FW leachate amount based on TN or by significantly reducing the nitrogen load in the subsequent wastewater treatment process. Moreover, the produced methanol can be an effective alternative for carbon source supply for denitrification in the subsequent process.

**Keywords:** digested food waste leachate; bio methanol production; denitrification; methanotrophs; biogas; wastewater treatment plant

---

## 1. Introduction

Since 2013 when the marine dumping of food waste (FW) leachate was prohibited in accordance with the London Convention on Prevention of Marine Pollution by Dumping of Wastes and Other Matter in 2009 (which was ratified in 2009), South Korea has moved to treatment in landfills and wastewater treatment plants [1,2]. Of the total FW leachates generated (9498 $m^3 \cdot day^{-1}$) in 2009, 4007 $m^3 \cdot day^{-1}$ was dumped in the sea. This extra load was diverted to land-based facilities such that FW leachates were treated together with other wastes due to the finite capacity of existing treatment plants. This resulted in a deterioration of the quality of the treated water, necessitating the expansion of treatment facilities and thus raising treatment costs [3].

Consequently, the number of biogas plants have increased to convert FWs and FW leachates to resources. Digested, non-biodegradable FW leachates generated from the biogas process are self-treated, mostly in private treatment facilities, whereas those generated in public treatment facilities are treated in urban wastewater treatment plants.

Through the Public Sewage Facility Operation Management Guidelines, the Ministry of Environment restricts the total nitrogen (TN) and total phosphorus (TP) load of FW leachate influent to wastewater treatment plants to within 10% of the influent sewage load design [4,5].

In parallel with government guidelines, each wastewater treatment plant sets independent influent water quality standards of FW leachate, considering the influent water quality and treatment process characteristics. Digested FW leachate has a low carbon-to-nitrogen (C/N) ratio due to high non-biodegradable chemical oxygen demand (COD) and ammonia concentrations. Thus, an additional carbon source is required to treat non-biodegradable FW leachates. It is important to identify an applicable carbon source because supplying an external carbon source for denitrification increases

wastewater treatment costs. Therefore, studies have been conducted to improve the economic efficiency using byproducts from anaerobic digestion processes (such as municipal solid wastes [6,7], FWs [8], and livestock wastes [9–12]) as carbon sources. The FW leachates discharged from FW biogas plants also require an appropriate carbon source due to the low C/N ratio (i.e., low carbon content and high nitrogen content), because $CH_4$ and $CO_2$ are discharged from the anaerobic degradation process. However, related studies are insufficient. In a previous study [13], we demonstrated that bio-methanol can be produced by supplementing $NaNO_3$ as a nitrogen source for methanotrophs in the final treated water of a wastewater treatment plant with sufficient trace elements and by using biogas as a carbon source.

To secure the sustainability of converting FWs into resources, this study aimed to reinforce the denitrification of the carbon source by producing bio-methanol using biogas that is self-produced in the anaerobic digester of the digested FW leachate and wastewater treatment plant. This leachate has a high NaCl content and is transported to and treated in a wastewater treatment plant from the biogas plant of FWs and FW leachates. Methanotrophs use the nitrogen in wastewater as a nitrogen source for growing and producing methanol by oxidizing methane while performing autotrophic denitrification [14]. The methanol produced by methanotrophs is the most widely used carbon source because its denitrification rate is higher than those of many other sources [15]. Furthermore, to contribute to improving the quality of wastewater treatment plant effluent (by reducing the nitrogen load through reducing the high ammonia nitrogen content in digestive fluid), this study examined bio-methanol production and simultaneous denitrification characteristics of urban wastewater treatment plants that perform connected treatment of digestive fluid generated from the FWs and FW leachate biogas plant.

## 2. Materials and Methods

### 2.1. Characteristics of the FW Leachate of Biogas Plant

The digested FW leachates generated from the biogas plant of the Goyang City Biomass Energy Facility are first treated in the plant and are then transported separately to the nearby Samsong Wastewater Treatment Plant and Ilsan Wastewater Treatment Plant for final treatment. This study only examined the digested FW leachate of the biogas plant for FWs and FW leachates flowing into the Ilsan Wastewater Treatment Plant, Goyang City, South Korea. The annual mean influent flow rate in 2018 was 234.6 ($\pm$36.9) $m^3$ $day^{-1}$, the mean pH was 8.0 ($\pm$0.5), the mean COD concentration was 279.1 ($\pm$116.6) $mg \cdot L^{-1}$, and the mean TN concentration was 252.2 ($\pm$99.2) $mg \cdot L^{-1}$ (Table A1).

### 2.2. Cultivation of Methanotrophs

Soil was collected from a depth of 15–20 cm from the cover layer soil at a Korean metropolitan landfill (coastal reclaimed land; 20,749,874 $m^2$; 37.57° N, 126.62° E; [Figure A1]) in Incheon, Korea. Soil samples were collected from three separate points and were uniformly mixed, filtered through a sieve (No. 50; 300 μm) and stored at 4 °C. To cultivate methanotrophs from the collected soil, a modified ammonia and nitrate mineral salt (ANMS) medium was prepared by mixing $NH_4Cl$ and $KNO_3$, which are nitrogen components of the ammonia mineral salt (AMS) and nitrate mineral salt (NMS) medium [16,17]. The medium contained the following (per L of distilled water): 0.5 g $KNO_3$; 0.25 g $NH_4Cl$; 1.0 g $MgSO_4 \cdot 7H_2O$; 0.2 g $CaCl_2 \cdot H_2O$; 0.1 mL 3.8% (w/v) Fe-EDTA solution; 0.5 mL 0.1% (w/v) $NaMo \cdot 4H_2O$; 26 g $KH_2PO_4$; 62 g $Na_2HPO_4 \cdot 7(H_2O)$. Additionally, 1 mL of trace element solution was added (per L of distilled solution: 500 mg $FeSO_4 \cdot 7H_2O$; 400 mg $ZnSO_4 \cdot 7H_2O$; 20 mg $MnCl_2 \cdot 7H_2O$; 50 mg $CoCl_2 \cdot 6H_2O$; 10 mg $NiCl_2 \cdot 6H_2O$; 15 mg $H_3BO_3$; 250 mg EDTA). All chemical reagents were of analytical grade and were purchased from Sigma-Aldrich (St. Louis, MO, USA).

Next, 200 mL modified ANMS medium was added to a 350 mL conical flask, and 5 g of the soil sample (prepared as described above) was added for microbial inoculation. The flask was closed on the open end with a silicon stopper and a methanol gas-tight syringe was used to add 20% (V%) methane to the upper 150 mL of the headspace. The flask was then sealed with parafilm and cultured

in a rotary shaker (Lab Champion IS-971R, Champion Laboratories, Albion, IL, USA). Following 24 h of spinner culture at 250 rpm at 30 °C, the cultured solution was left to settle for 10 min, after which 100 mL of the supernatant was removed and added to a new 350 mL conical flask containing NMS medium. The procedure was repeated four times, and the resulting solution containing NMS medium was finally separated using a centrifuge (Dongseo Science Ltd., Dangjin, Centrifuge-416 Korea) at 2700× *g*. The centrifuged pellet was freeze-dried at −55 °C in a freeze dryer (OPERN FDS-12003, Seoul, Korea) for subsequent use. (The steps are summarized in Table A2).

*2.3. Biogas*

To maintain the consistency of the experiment, the biogas was simulated and synthesized to reflect the characteristics of biogas from the digester of a wastewater treatment plant. Biogas consisted of $CH_4$ (67.0%), $CO_2$ (31.0%), $N_2$ (1.3%), and $O_2$ (0.7%).

*2.4. Analysis and Measurements*

2.4.1. Analysis of Microbial Community

For microbiological analysis, DNA extraction, PCR (Polymerase chain reaction) amplification, and pyrosequencing were performed by Chunlab, Inc. (Seoul, Korea). The 16S rRNA genes of each sample were amplified using barcoded universal primers. To compare each sample's operational taxonomic units, shared operational taxonomic units were obtained by XOR analysis with the CLcommunity program (Chunlab, Inc.). The composition and ratio of microbial species shared by the three sets of samples were calculated (Table A3).

2.4.2. Analysis of Biogas and Microbial Metabolites

Biogas and metabolite measurements, microbial community analysis, and water quality analysis followed the methods described in a previous study [13].

2.4.3. Analysis of Water Quality and Organic Biodegradability

Analyses of water quality were based on the American Standard Methods for the Examination of Water and Wastewater (23rd ed., American Public Health Association) and EPA Methods (EPA Method 1613). Organic components were analyzed by the colorimetric method and the atomic analysis was performed by atomic-absorption-spectroscopy. The COD fractions method [18,19] was used to evaluate the biodegradability of organic matter.

*2.5. Batch Testing and Assay Device for Anoxic-aerobic and Sequencing Batch Reactor Process*

For the batch test, the freeze-dried cells (described above) were inoculated in a 160 mL serum bottle (Sigma-Aldrich, USA) with 50 mL of digested FW leachate (collected from the modified ANMS medium or wastewater treatment plant) as the culture solution to prepare 550 mg·L$^{-1}$ based on the mixed liquor suspended solids (MLSS). Furthermore, in the batch test, 25 mL (or 22.7%) of the headspace (110 mL) of the 160 mL serum bottle was replaced with biogas to adjust the $O_2$:$CH_4$ ratio to 17.3 mL:18.6%; the same ratio was maintained in the following experiments. Then, the samples were collected and analyzed while being cultured in a shaking incubator at 30 °C at 150 rpm.

The anoxic-aerobic process (Ludzack-Ettinger process [20]) in Figure 1 was used to examine the typical denitrification characteristics of the wastewater. The operating conditions are shown in Table 1 [20]. After measuring the amount of nitrate-nitrogen ($NO_3^-$-N) formed in the aerobic tank when methanol was injected as a carbon source for denitrification, 3.0 g methanol per 1 g of $NO_3^-$-N was injected, based on the empirical reaction equation and experimental value (Table A4) as suggested by McCart et al. [21].

Figure 1 image:

**Figure 1.** Anoxic-aerobic denitrification (Ludzack-Ettinger) process (pre-denitrification).

**Table 1.** Operation conditions of the anoxic-aerobic (Ludzack-Ettinger) process.

| Item | MLSS (mg·L$^{-1}$) | Inflow (L·day$^{-1}$) | HRT (h) |
|------|--------------------|------------------------|---------|
| Oxic | 2200-2300 | 24 | 4.3 |
| Anoxic | 2300-2400 | | 1.7 |

Note: Internal recycle: 3Q, HRT—hydraulic retention time, MLSS—mixed liquor suspended solids.

The sequencing batch reactor (SBR) was equipped with a stirrer, a gas supplier, and gas pressure gauges as shown in Figure 2. The SBR was operated with a residence time of 6 h while stirring at 20–25 °C at 150 rpm, and with the settling time of 30 min. The microbes increased by proliferation and were discharged with a sludge retention time (SRT) of 15 days, while the MLSS was maintained at 3700–4400 mg·L$^{-1}$. The biogas was automatically injected from the gas tank while monitoring biogas and oxygen consumption by the pressure sensor in the bioreactor.

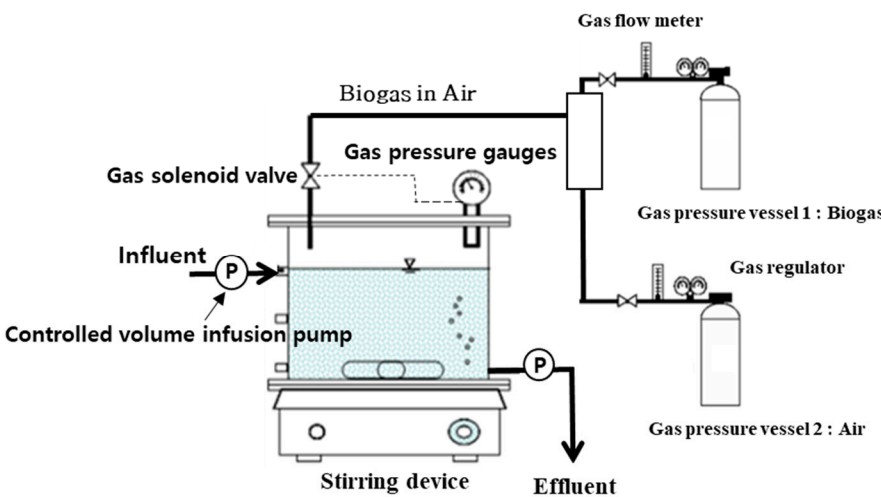

**Figure 2.** Schematic diagram of the methane and methanol dependent microbial bacteria consortium sequencing batch reactor (SBR) system.

## 3. Results and Discussion

### 3.1. Water Quality and Biological Treatment Characteristics of FW Leachate

The FWs and FW leachates flowing into the FW biogas plant progress through the anaerobic digestion process were composed of; (1) leachate (including organic matter), (2) biogas, and (3) stabilized digestion sludge, before being discharged from the digester. Digested FW leachate cannot be directly discharged due to the high residual organic matter concentration. Thus, it undergoes aerobic treatment or secondary treatment (such as advanced oxidation process [AOP]) before being transported

to the wastewater treatment plant for retreatment in accordance with the water quality standards for public waters and effluent. The Ilsan Wastewater Treatment Plant only accepts carried-in digested FW leachate in which the nitrogen load has been treated to below 7 kg·day$^{-1}$ or 300 m$^3$·day$^{-1}$ by applying the TN criterion. This is because it is difficult to treat nitrogen due to the water quality characteristics of the influent and the treatment process. As shown in Table 2, the TN concentration was 297 mg·L$^{-1}$ at pH 8.1, which is higher than the average (annual mean value) concentration in sewage influent of 53.8 ($\pm$13.0) mg·L$^{-1}$, and the NH$_4$-N concentration was 236 mg·L$^{-1}$, accounting for the majority of the TN. Furthermore, the influent digested FW leachate had a very high NaCl content of 0.73%, unlike other wastewater types.

**Table 2.** Water quality characteristics of food waste (FW) leachate, digested FW leachate after conversion to biogas, and FW leachate carried in the wastewater treatment plant. SS—suspended solids, COD—chemical oxygen demand, TN—total nitrogen, TP—total phosphorus, WC—water content, STP—sewage treatment plant.

| | pH | SS (mg·L$^{-1}$) | COD (mg·L$^{-1}$) | NH$_4$-N (mg·L$^{-1}$) | TN (mg·L$^{-1}$) | TP (mg·L$^{-1}$) | NaCl (%) | WC (%) |
|---|---|---|---|---|---|---|---|---|
| FW leachate | 4.2 ($\pm$0.13) | 98,457 ($\pm$239.32) | 148,605 ($\pm$612.4) | 648 ($\pm$23.6) | 3729 ($\pm$116.2) | 428 ($\pm$31.2) | 0.52 ($\pm$0.12) | 89.6 ($\pm$3.2) |
| Digested leachate | 8.2 ($\pm$0.11) | 56,032 ($\pm$311.3) | 44,289 ($\pm$276.4) | 1823 ($\pm$87.5) | 3654 ($\pm$98.5) | 317 ($\pm$24.3) | 0.95 ($\pm$0.07) | 87.3 ($\pm$2.4) |
| Carried in STP | 8.1 ($\pm$0.08) | 230 ($\pm$13.7) | 279 ($\pm$29.7) | 236 ($\pm$18.6) | 297 ($\pm$21.4) | 5 ($\pm$1.1) | 0.73 ($\pm$0.09) | 89.7 ($\pm$2.6) |

Note: The NaCl value was converted from the measured concentration of chlorine ions, using the following formula: salinity (%) = 0.00018066 5 Cl$^-$ (mg·L$^{-1}$).

Organic matter that is easily degradable through anaerobic digestion was mostly degraded and the soluble biodegradable chemical oxygen demand (BDCOD) and non-biodegradable chemical oxygen demand (NBDCOD) in the influent digested FW leachate had concentrations of <1%. The total BDCOD of the influent digested FW leachate in the wastewater treatment plant accounted for 28.7% of the COD, and NBDCOD accounted for the remaining 71.3% of the COD (Table 3). It can be seen that the majority of these FW leachates are difficult to remove by biological treatment methods in the wastewater treatment process and have components that are disadvantageous for subsequent water treatment and effluent.

**Table 3.** Analysis of water quality characteristics by biological treatment of leachate by the food waste (FW) leachate treatment step (unit: %). BDCOD—biodegradable chemical oxygen demand, NBDCOD—non-biodegradable chemical oxygen demand.

| | Soluble BDCOD | Soluble NBDCOD | Particle BDCOD | Particle NBDCOD | Total BDCOD | Total NBDCOD |
|---|---|---|---|---|---|---|
| FW leachate | 22.4 | 6.4 | 39.5 | 31.7 | 61.9 | 38.1 |
| Digested leachate | 0.6 | 0.3 | 33.3 | 65.8 | 33.9 | 65.8 |
| Carried in STP | 0.9 | 0.7 | 27.8 | 70.6 | 28.7 | 71.3 |

Note: Data represent means from two replicates.

When the denitrification experiment was conducted using the anoxic/aerobic denitrification process for the influent digested FW leachate of the wastewater treatment plant showing these characteristics, only 12.7% of the influent TN was removed, as shown in Table 4. When methanol was input to the anoxic process as a carbon source required in correspondence to the additionally generated nitrate (NO$_3$$^-$-N) produced by nitrification, the TN removal rate increased to 39.1%. The above results of food waste leachate analysis show that the organic acid in anaerobically digested FW leachates, which is available as a carbon source, is mostly degraded and composed of non-biodegradable organic matter,

thus raising the pH of the surrounding area to eight and necessitating further chemical treatments. Therefore, this negative environmental impact makes the process less sustainable as a method for generating resources from biogas.

**Table 4.** Denitrification characteristics of the influent digested food waste (FW) leachate in the wastewater treatment plant (mean values of 38 day operation results).

| Item | Influent (mg/L) | Anoxic/Aerobic Process | | Methanol Addition in Anoxic Reactor | |
|---|---|---|---|---|---|
| | | Effluent conc. (mg/L) | Removal (%) | Effluent Conc. (mg/L) | Removal (%) |
| COD | 263.4 (±23.7) | 205.5 (±16.5) | 22.0 (±3.4) | 198.6 (±16.7) | 24.6 (±2.8) |
| NH$_3$-N | 228.2 (±19.3) | 123.4 (±18.8) | 45.9 (±4.2) | 126.3 (±14.5) | 98.9 (±4.1) |
| NO$_3{}^-$-N | 71.6 (±9.4) | 142.6 (±20.3) | 12.7 as T-N (±3.6) | 63.5 (±9.3) | 39.1 as T-N (±4.6) |
| PO$_4{}^{-3}$-P | 7.6 (±8.2) | 4.8 (±3.2) | 36.8 (±4.1) | 3.4 (±1.6) | 5.3 (±1.7) |

*3.2. Change in the Characteristics of the Microbial Community in the Culture Medium*

When the methanotrophs were cultured using the digested FW leachate with the modified ANMS described in 2.2 and the characteristics described in 3.1 as culture sources, the *Methylomonas, Methylococcus, Methylobacter, Methylomonas_f_uc,* and *Methylosarcina* genera were cultured in the landfill soil as shown in Figure 3.

In the case of the methanotrophs cultured in the first modified ANMS, *Methylomonas methanica* was dominant (36.9%), followed by *Methylomonas* sp. (18.1%), and *Methylobacter marinus* (12.3%). In contrast, when the methanotrophs in the modified ANMS were cultured using the digested FW leachate as the culture solution, the *Methylomonas* genus did not appear, and *Methylobacter marinus* (which are marine methane-oxidizing bacteria) accounted for most of the constituent species at 88.2% of methanotrophs [22,23]. The reason that *Methylobacter marinus* was detected as a dominant species in this study seemed to be that the urban waste landfill (which was the soil sampling point for seeding) is reclaimed marine land, and the NaCl concentration of the digested FW leachate was 0.52–0.95% higher than that of other wastewater (Table 2).

According to Bowman et al. and Wartiainen et al., the salinity range for optimal growth was wide (0.0–3.0% NaCl) [23–25]. Therefore, it was verified that the *Methylobacter marinus* cultivated as a dominant species in this study can grow in a wide range of salinities and play the role of a dominant species among methanotrophs when FW leachate with a high NaCl content is used.

*Methylobacter marinus* is a marine methane-oxidizing bacterium that commonly appears in other studies on marine methanotrophs [25,26]. No further studies have been conducted on this species and no clear taxonomic status has been found until now. Recently, Flynn et al. found the genome sequence of *Methylobacter marinus* A45 [27]. Furthermore, it is presumed that the species composition was simplified to the unequalled dominance of *Methylobacter marinus* because NaCl suppresses the expression of methanol dehydrogenase (MDH), which oxidizes methanol generated by methanotrophs to formaldehyde, thus inhibiting the formaldehyde metabolism of methanotrophs which do not have a wide-ranging tolerance to NaCl. A possible explanation for this may be that NaCl acts as a MDH expression inhibitor to prevent transformation of methanol into formaldehyde during methanol production using methanotrophs [28–33].

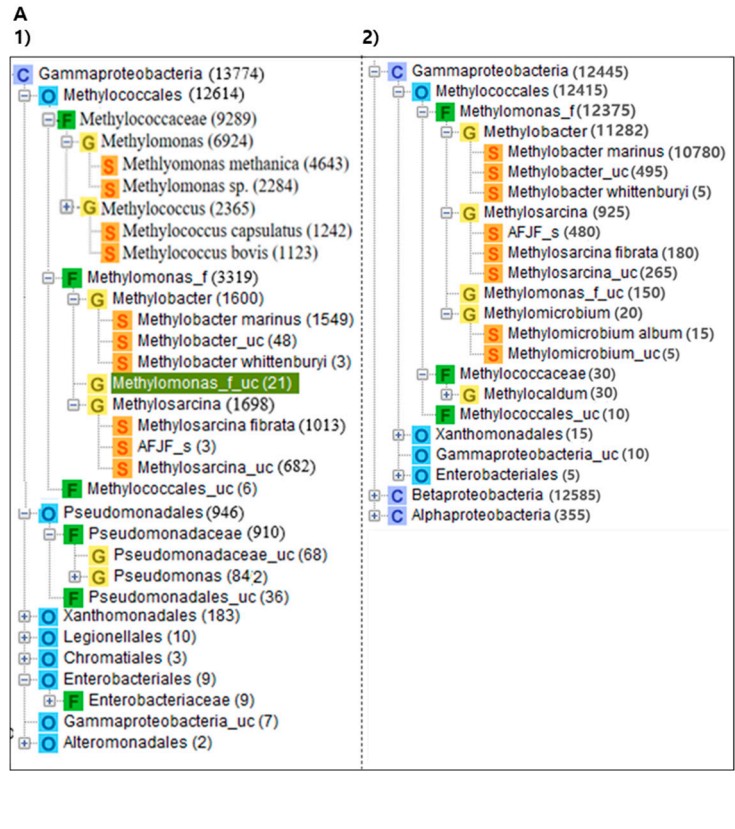

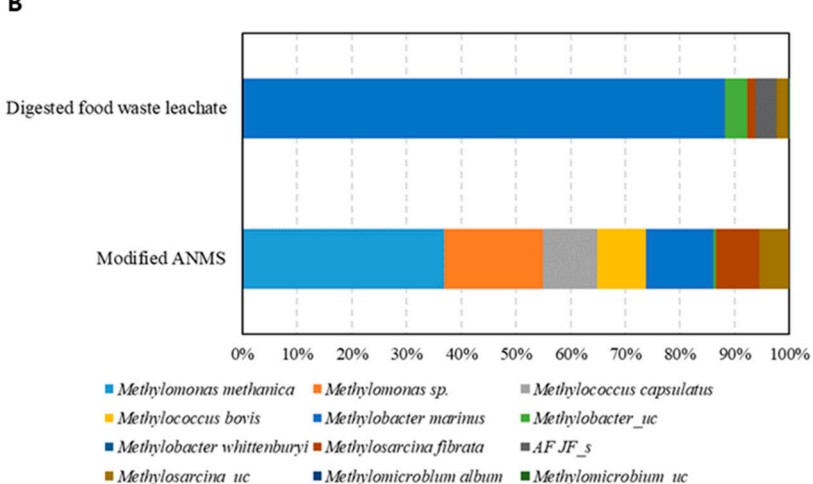

**Figure 3.** Schematic diagram of the constituent species of methanotrophs from the; (**A**) (1) modified ammonia and nitrate mineral salt (ANMS) medium and (2) digested food waste (FW) leachate, and (**B**) the relative ratios of constituent methanotrophs species.

### 3.3. Methanol Production from Digested FW Leachate by Methanotrophs

Table 5 and Figure 4 outlines the methanol and formaldehyde production from methanotrophs and biogas based on the modified ANMS medium and digested FW leachate. After 24 h culturing in a LMSS concentration of 550 mg·L$^{-1}$, methanol was produced at 4.11 mM (COD converted value: 197.54 mg·L$^{-1}$) and 3.82 mM (COD converted value 131.90 mg·L$^{-1}$) in the modified ANMS medium and the digested FW leachate, respectively. The digested FW leachate produced somewhat less methanol from methanotrophs than the modified ANMS medium. The ratio of methanol in the total metabolites was 80.3%, which was higher than the modified ANMS medium (60.3%). Thus, the digested FW leachate had only a minimal disadvantage compared to the modified ANMS medium in terms of methanol formation.

**Table 5.** Production of methanotrophs metabolite (methanol, formaldehyde, formate) in modified ammonia and nitrate mineral salt (ANMS) medium and digested food waste (FW) leachate (for 24 h).

| Culture Medium | Methanol (mM) | Formaldehyde (mM) | Formate (mM) | Total Metabolite (mM) |
|---|---|---|---|---|
| Modified ANMS medium | 4.11 (±0.13) | 1.98 (±0.12) | 0.73 (±0.08) | 6.82 (±0.19) |
| Digested FW leachate | 3.82 (±0.17) | 0.81 (±0.09) | 0.13 (±0.02) | 4.76 (±0.14) |

Note: Data represents means from three replicates ± standard deviations.

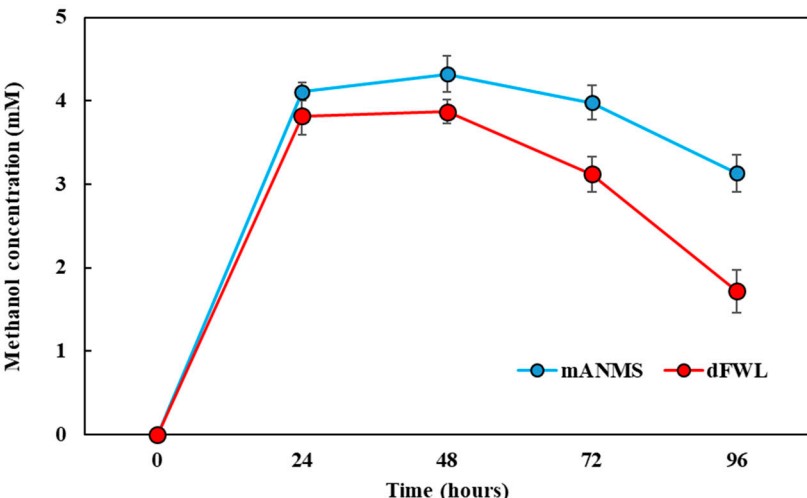

**Figure 4.** Temporal changes in methanol production from methanotrophs with modified ammonia and nitrate mineral salt (ANMS) medium (mANMS) and digested food waste (FW) leachate (dFWL) as culture solutions. Data represent means from three replicates ± standard deviations.

## 3.4. Denitrification Characteristics of Digested FW Leachate

Anthony [34] revealed experimentally that methane-oxidizing bacteria have relatively high nitrogen demand during growth and ingested 0.25 mol of nitrogen to assimilate 1 mol of carbon from methane. Furthermore, the nitrogen and phosphorus contents in the methanotroph sludge showed that the TN content in the activated sludge of the wastewater treatment plant was 5.34% on average, but was 8.53% in the methane-oxidizing bacteria sludge. This result suggests that methanotroph growth has a high nitrogen demand [35].

Furthermore, in the metabolism process of methanotrophs, ammonia was oxidized to $N_2O$ through $NH_3OH$ in the same path as that of the ammonia oxidation of ammonia-oxidizing bacteria (AOB) by the methane monooxygenase (MMO), which is a methane-oxidizing enzyme [34,35]. Consequently, the tendency of removing high concentration ammonium and some nitrates in the digested FW leachate by using methanotrophs and biogas was verified. As shown in Figure 5, there were decreases in ammonia by 73.2% (from 276 mg·$L^{-1}$ to 74 mg·$L^{-1}$), nitrate by 61.0% (from 82 mg·$L^{-1}$ to 32 mg·$L^{-1}$), and TN by 67.4% (from 374 mg·$L^{-1}$ to 122 mg·$L^{-1}$) in 6 h. In other words, the method applied to this experiment was verified as an effective means to increase methanol as a carbon source for the denitrification of digested FW leachate with low biodegradability and high NaCl concentration, and also for the denitrification of the subsequent sewage treatment process.

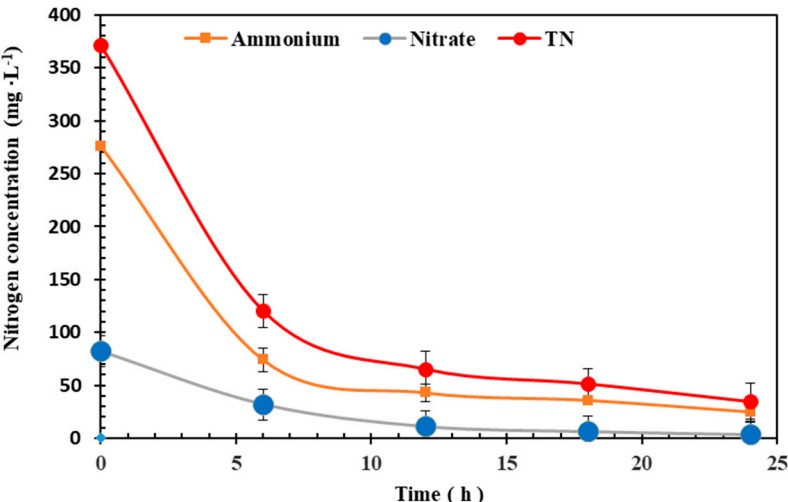

**Figure 5.** Changes in nitrogen concentrations in digested food waste (FW) leachate with methanotrophs and biogas. TN—total nitrogen. Note: Data represents means from three replicates ± standard deviations.

Based on the denitrification characteristics revealed through the batch test, the characteristics of denitrification from the digested FW leachate were analyzed by continuously operating the SBR system (Figure 1) for 37 d (based on 6 h operation per session), with the digested FW leachate flowing into the wastewater treatment plant. The results are shown in Figure 6.

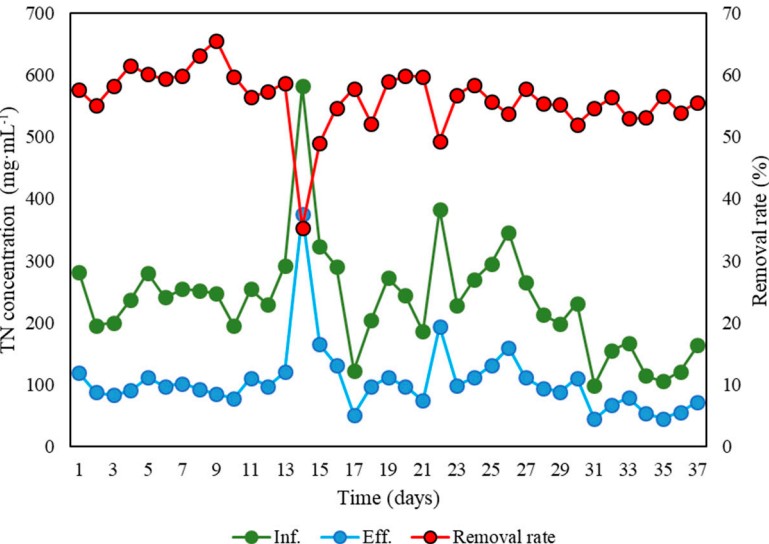

**Figure 6.** Results of the continuous experiment of SBR (operation time: 6 h) for the denitrification characteristics of digested food waste (FW) leachate by methanotrophs and biogas. Inf.—influent, Eff.—effluent, TN—total nitrogen.

Based on TN, the mean influent concentration was 236.4 (±87.2) mg L$^{-1}$, the mean effluent concentration was 105.7 (±56.1) mg·L$^{-1}$, and the mean removal rate was 56.1% (±4.9). Hence, a much higher TN removal rate than the 39.1% (Table 4) in 3.1 when methanol was additionally supplied to the anoxic/aerobic denitrification process was obtained. Therefore, the proposed method can contribute to improving the treatment efficiency by accommodating twice the current influent FW leachate amount (based on TN) or by significantly reducing the nitrogen load in the subsequent wastewater treatment process. Moreover, it can be an alternative to improve the denitrification efficiency by adding the

generated methanol to the influent water as a carbon source for denitrification. In this case, the removal rate decreased when ammonium—which accounts for most of the TN component—increased above a certain level. This was presumably caused by two toxic intermediates of ammonium oxidation by methanotrophs, hydroxylamine ($NH_2OH$) and nitrite, interfering with the normal operation of the detoxification mechanisms, or by complex causes arising from the energy consumed by methanotrophs during the detoxification of hydroxylamine [36–38].

Further studies are required to investigate the primary treatment method for high-concentration ammonia wastewater in biogas plants, which are the source as well as the methanol production metabolism characteristics for methanotrophs with a high NaCl tolerance.

## 4. Conclusions

Bio-methanol was produced at 3.82 mM (COD; 131.90 mg·$L^{-1}$) from FW leachates having few biologically available carbon sources, rich nitrogen sources (AMS; $NH_4$-N 168.35 mg $L^{-1}$, NMS; $NO_3^-$-N 613 mg·$L^{-1}$), and an mean content of 0.73% or higher of digested NaCl (such as methanotroph culture medium), by using biogas with methylobacter *(M. marinus)* as the dominant methanotroph species. Furthermore, up to 56.1% of the TN in the FW leachates could be removed by oxidizing ammonia through nitrogen assimilation of methanotrophs and methane monooxygenase (MMO). Therefore, the proposed method can contribute to improving the treatment efficiency by accommodating twice the current influent FW leachate amount based on TN or by significantly reducing the nitrogen load in the subsequent wastewater treatment process. Moreover, it can be a new alternative for the sustainable operation of biogas plants to improve the denitrification efficiency by adding the generated methanol to the influent water as a carbon source for denitrification.

**Funding:** This work was supported by a major project of the Korea Institute of Civil Engineering and Building Technology, grant number 20190152-001

**Acknowledgments:** I would like to thank Sungkyunkwan University Center for zero emission technology (https://enc.skku.edu/enc/lab/environ_tech.do), and analysis of metabolites and water quality, ©CHUNLAB (chunlab.com) for support of bioinfomatic information for microbiology and Editage (www.editage.co.kr) for English language editing.

**Conflicts of Interest:** The authors declare no conflicts of interest.

## Appendix A

## Tables and Figures

**Simultaneous Denitrification and Bio-methanol Production for Sustainable Operation of Biogas Plants**

**Table A1.** Influent amount of digested FW leachate and water quality for target biogas plant in the wastewater treatment plant. BOD—Biological oxygen demand, COD—Chemical oxygen demand, SS—suspended solids, TN—total nitrogen, TP—total phosphorus.

|  | Flow | Water Temp. | PH | BOD | COD | SS | TN | TP |
|---|---|---|---|---|---|---|---|---|
| Mean | 234.6 | 14.6 | 8.0 | 171.5 | 279.1 | 245.3 | 252.2 | 4.9 |
| N | 302.0 | 43.0 | 43.0 | 43.0 | 43.0 | 43.0 | 43.0 | 42.0 |
| SD | 36.9 | 6.8 | 0.5 | 99.1 | 116.6 | 134.6 | 99.2 | 3.5 |

**Table A2.** Methanotrophs enrichment procedure.

1.  Collect soil from below 15 cm depth in the waste landfill site
2.  Sieve (#50) collected soil to remove debris
3.  5 g soil added to 200 mL NMS medium in 350 mL flask
4.  Capped and sealed
5.  Inject methane into the headspace (150 mL) of the flask (20% $CH_4$ by volume)
6.  Incubate for 1 d at 250 rpm and 30 °C
7.  Settle for 10 min and decant the supernatant
8.  Add 100 mL supernatant to 100 mL of NMS medium in a 350 mL flask
9.  Conduct steps (4)–(7)
10. Repeat steps (8)–(9) four times
11. Centrifuge the supernatant for 5 min at 2,700 $\times g$
12. Freeze and dry of centrifuged solids at −55 °C and store in refrigerator at 4 °C for subsequent use.

**Table A3.** Conditions used in touch-down Image result for polymerase chain reaction (PCR) for microbial community analysis.

| Step | Temperature (°C) | Time (s) | Cycle |
|---|---|---|---|
| Initial denaturation | 94 | 300 | – |
| Denaturation | 94 | 30 | 10 |
| Annealing | 60 | 45 | (−0.5 °C/cycle) |
| Extension | 72 | 90 | |
| Denaturation | 94 | 30 | |
| Annealing | 55 | 45 | 20 |
| Extension | 72 | 90 | |
| Hold | 4 | ∞ | – |

**Table A4.** Basis of calculating the methanol demand required for nitrate denitrification [a].

McCarty et al. (1969) experimentally measured the cell production using methanol as carbon source and established the empirical reaction equations of the denitrification process as follows:

$$NO_3^- + 1.08CH_3OH + 0.24H_2CO_3 \rightarrow 0.056C_5H_7O_2N + 0.47N_2 + 1.68H_2 + HCO_3^- \tag{1}$$

$$NO_2^- + 0.67CH_3OH + 0.53H_2CO_3 \rightarrow 0.04C_5H_7O_2N + 0.48N_2 + 1.23H_2 + HCO_3^- \tag{2}$$

$$O_2 + 0.93CH_3OH + 0.56NO_3^- \rightarrow 0.056C_5H_7O_2N + 1.04H_2O + 0.59H_2CO_3 + 0.56HCO_3^- \tag{3}$$

In Equation (1), the theoretical methanol amount required to remove 1 g of nitrate nitrogen is approximately 2.47 g, and the experimental value was 2.5–3.0 g/g $NO_3^-$-N.

[a] McCarty, Perry L., Beck, Louis, St. Amant, Percy, Biological denitrification of wastewaters by addition of organic materials, Proceedings of the 24th Industrial Waste Conference, Engineering Technical Reports Collection, Purdue University, 1969. 1271-1285. http://earchives.lib.purdue.edu/u?/engext,16392.

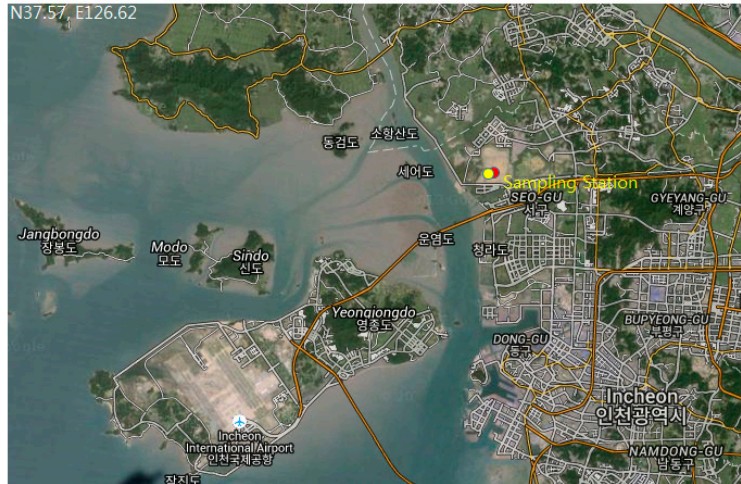

**Figure A1.** Location of soil sample collection in the metropolitan landfill (coastal reclaimed land, 20,749,874 m$^2$).

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
