# Peer review of "Simultaneous Denitrification and Bio-Methanol Production for Sustainable Operation of Biogas Plants"

_sustainability, doi:10.3390/su11236658_

Round 1
Reviewer 1 Report
The presented paper deals with an important environmental issue connected with improving the water quality of wastewater treatment plant effluents. The Author proposes to reduce the nitrogen load of the treatment process through nitrogen reduction while producing bio-methanol as a denitrified carbon source using digested food waste leachate carried into wastewater treatment plants. The methodological part of the presented results and their discussion is thorouh and well supported and might be of potential interest for the Sustainability readers.
Nevertheless taking into account the range of analyses and the scope of the experiment it is surprising that all the work was done by just one person. Please address that issue.
Furthermore the papr is well-written and there are only few improvements I suggest to consider to make some paper fragments easier to follow. In the Abstract L 12-15: please simplify the sentence - although it summarizes the most imortant achievement - it is too long and hence difficult to follow.
In the introduction part (Line 32) please check the logic of the sentence- what "adverse effect in the search..." does Author mean?
Author Response
Thank you for your suggestion. To facilitate your review of our revisions, the following is a response to the questions and comments delivered in your letter dated 02 Nov 2019.
Point 1: Nevertheless taking into account the range of analyses and the scope of the experiment it is surprising that all the work was done by just one person. Please address that issue.
Response 1: This work was supported by a Major Project of the KICT (Korea Institute of Civil Engineering and Building Technology), grant number 20190152-001. This organization conducts national R&D projects funded by the South Korean government. As a researcher of this organization, I designed the research of a specific part and directly carried out the whole process. This study requires analysis of DNA and RNA related to microorganisms and analysis of enzymes and microbial metabolites expressed by them. Precision instrument analysis is also essential for this study. These analytical instruments and techniques are wide-ranging and cannot be performed by any one organization. Therefore, I took charge of the experimental design, operation of the experimental facilities, data analysis, and derivation of results. In-depth analyses in a related specialist field was outsourced to an institution specializing in this field. This organization is a for-profit entity and the analysts who performed the analyses on samples originating from the author's organization (KICT) are anonymous employees of the organization, not researchers.
Furthermore, this manuscript was written in Korean, and Korean translators and native editors who majored in related fields participated in the rewriting of this paper in English. The author and his affiliated organization believe that it is impossible in terms of research ethics to include employees of an anonymous commercial institution, who did not directly contribute to the results of this study, as authors. Therefore, the author mentioned and thanked the institutions that contributed to the preparation of this paper in the Acknowledgement. I hope this explains how I, as a single author could conduct an analytical study of this broad scope.
Acknowledgments: I would like to thank SUNGKYUNKWAN UNIVERSITY CENTER FOR ZERO EMISSION TECHNOLOGY (https://enc.skku.edu/enc/lab/environ_tech.do) for analyses of metabolites and water quality, â“’CHUNLAB (chunlab.com) for support with bioinformatic information of microbiology, Editage (www.editage.co.kr) for English language editing.
Point 2: I suggest to consider to make some paper fragments easier to follow. In the Abstract L 12-15: please simplify the sentence - although it summarizes the most important achievement - it is too long and hence difficult to follow.
Response 2: As the reviewer pointed out, I also think that the sentence is complicated and difficult to understand. Therefore, according to the author's guide, I shortened the sentence in Lines 12-15 to express only the core information as follows:
The aim of this study is to reduce the nitrogen load of the treatment process while producing bio-methanol using digested FW leachate diverted into wastewater treatment plants.
Point 3: In the introduction part (Line 32) please check the logic of the sentence- what "adverse effect in the search..." does Author mean?
Response 3: I intended to explain that South Korea had discarded enormous amounts of food waste (FW) leachates by marine dumping. To comply with “London Convention on Prevention of Marine Pollution by Dumping of Wastes and Other Matter in 2009,” all the FW leachates are now processed on the land; however, they are carried into wastewater treatment plants due to insufficient treatment facilities. The sentence explains that this is causing the problem of water quality deterioration by combined treatments and the expansion of treatment facilities, thus raising treatment costs.
We have revised this sentence as follows.
Of the total FW leachates generated (9,498 m3·day-1) in 2009, 4,007 m3·day-1 was dumped in the sea. This extra load was diverted to land-based facilities such that FW leachates were treated together with other wastes due to the finite capacity of existing treatment plants. This resulted in a deterioration of the quality of the treated water, necessitating the expansion of treatment facilities and thus raising treatment costs [3]
Again, thank you for giving us the opportunity to strengthen our manuscript with your valuable comments and queries. We have worked hard to incorporate your feedback and hope that these revisions persuade you to accept our submission.
Sincerely,

Reviewer 2 Report
This manuscript proposed a method aiming to improve the wastewater treatment efficiency utilizing methanol from biogas as carbon source for denitrification. The author also discusses the performance of methanol productions based on different mediums. The result shows that over 50% of total nitrogen removal rate can be obtained in average. The method using biogas as carbon source is relevant to Sustainability.
It is a complete manuscript and is acceptable after some questions are clarified.
Several studies have applied methane from biogas as a feasible carbon source for denitrification while methanol is chosen as sole carbon source in this manuscript. Please give a description or discussion of the reasons choosing methanol as the carbon source rather than methane or others. Food waste leachate as external carbon source for some practical applications in wastewater treatment have drawbacks like consumption of large amount of chemicals for pH adjustment. With regard to “sustainable” in this manuscript, it is important to identify whether the additional process for the obtainment of carbon is of positive impact on environment or finance. Please have a specific description of sustainability. In Fig. 6, decrease in removal rates along with increase of influent total nitrogen concentration is presented. It seems that the removal rate is unstable based on this figure. Explanation of this phenomenon is necessary.
Author Response
Thank you for your suggestion. To facilitate your review of our revisions, the following is a response to the questions and comments delivered in your letter dated 02 Nov 2019.
Point 1: Several studies have applied methane from biogas as a feasible carbon source for denitrification while methanol is chosen as sole carbon source in this manuscript. Please give a description or discussion of the reasons choosing methanol as the carbon source rather than methane or others.
Response 1: As the Reviewer pointed out, the explanation about selecting methanol produced by methanotrophs as a carbon source was insufficient. Therefore, we have added references to support this.
Methanotrophs use the nitrogen in wastewater as a nitrogen source for growth and produce methanol by oxidizing methane while performing autotrophic denitrification [14]. The methanol produced by methanotrophs is the most widely used carbon source because its denitrification rate is higher than those of many other sources [15].
14 Mechsner, K.L.; Hamer, G. Denitrification by Methanotrophic, Methylotrophic Bacterial Associations in Aquatic Environments. Plenum Press. 1983, 257-271.
15 Tam, Y.F.; Wong, Y.S. Effect of Exogenous Carbon Sources on Removal of Inorganic Nutrient By Nitrification-Denitrification Process. Water Res. 1992, 26, 1229–1236.
Point 2: Food waste leachate as external carbon source for some practical applications in wastewater treatment have drawbacks like consumption of large amount of chemicals for pH adjustment. With regard to “sustainable” in this manuscript, it is important to identify whether the additional process for the obtainment of carbon is of positive impact on environment or finance. Please have a specific description of sustainability.
Response 2: As the reviewer pointed out, FW leachates contain organic acids, which cannot be used as a carbon source. However, the anaerobic-digested FW leachates are mostly composed of non-biodegradable organic matters because almost all the organic matter available as a carbon source have been degraded, and the pH of its surrounding are also increases to 8. Therefore, we have added the following explanation in 3.1. Water quality and biological treatment characteristics of FW leachate.
The above results of food waste leachate analysis show that the organic acid in anaerobically digested FW leachates, which is available as a carbon source, is mostly degraded and composed of non-biodegradable organic matter, thus raising the pH of the surrounding area to 8 and necessitating further chemical treatments. Therefore, this negative environmental impact makes the process less sustainable as a method for generating resources from biogas.
Point 3: In Fig. 6, decrease in removal rates along with increase of influent total nitrogen concentration is presented. It seems that the removal rate is unstable based on this figure. Explanation of this phenomenon is necessary.
Response 3: We have examined related literature regarding this phenomenon. We found a study reporting that the intermediaries formed by methanotrophs at high concentrations of ammonium, which accounts for most of the TN, act as toxic substances, thereby reducing the activity of methanotrophs. Thus, we have added the relevant reference and provided a supplementary explanation of the reason.
In this case, the removal rate decreased when ammonium—which accounts for most of the TN component—increased above a certain level. This was presumably caused by two toxic intermediates of ammonium oxidation by methanotrophs, hydroxylamine (NH2OH) and nitrite, interfering with the normal operation of the detoxification mechanisms, or by complex causes arising from the energy consumed by methanotrophs during the detoxification of hydroxylamine [38].
Zheng, Y.; Huang, R.; Wang, B.Z.; Bodelier, P.L.E.; Jia, Z.J. Competitive interactions between methane- and ammonia-oxidizing bacteria modulate carbon and nitrogen cycling in paddy soil. Biogeosciences 2014, 11, 3353–3368.
Again, thank you for giving us the opportunity to strengthen our manuscript with your valuable comments and queries. We have worked hard to incorporate your feedback and hope that these revisions persuade you to accept our submission.
Sincerely,

Round 2
Reviewer 2 Report
I have checked the whole manuscript again and I am grateful for the author's supplement to this manuscript.
The author is able to find studies and papers to support this research and it seems more complete than previous version.
I have just found one small mistake that there are two section 3.1 in this manuscript. An overall review of the form is suggested.
Author Response
Point
I have just found one small mistake that there are two section 3.1 in this manuscript. An overall review of the form is suggested.
Response
Thank you for your reconsideration the whole manuscript again. To facilitate your second review of our revisions, the following is a response to the questions and comments delivered in your letter dated 16 Nov 2019.
Your review has helped us improve the quality of this manuscript. So this is a logical improvement over the previous version. Thank you very much for this.
The duplicate of 3.1 you pointed out is our mistake. We removed the duplicate 3.1 from the manuscript and added 3.2
Again, thank you for giving us the opportunity to strengthen our manuscript with your valuable comments and queries.
Sincerely,